# Performance of TiO_2_-Based Tubular Membranes in the Photocatalytic Degradation of Organic Compounds

**DOI:** 10.3390/membranes13040448

**Published:** 2023-04-20

**Authors:** Carmen Barquín, Aranza Vital-Grappin, Izumi Kumakiri, Nazely Diban, Maria J. Rivero, Ane Urtiaga, Inmaculada Ortiz

**Affiliations:** 1Departamento de Ingenierías Química y Biomolecular, Universidad de Cantabria, Avda. Los Castros s/n, 39005 Santander, Spain; carmen.barquindiez@unican.es (C.B.); mariajose.rivero@unican.es (M.J.R.);; 2Graduate School of Science and Technology for Innovation, Graduate School Science and Engineering, Yamaguchi University, Ube 755-8611, Japan; izumi.k@yamaguchi-u.ac.jp

**Keywords:** photocatalysis, filtration, membrane, TiO_2_, TiO_2_/Ag

## Abstract

This work presents the photocatalytic degradation of organic pollutants in water with TiO_2_ and TiO_2_/Ag membranes prepared by immobilising photocatalysts on ceramic porous tubular supports. The permeation capacity of TiO_2_ and TiO_2_/Ag membranes was checked before the photocatalytic application, showing high water fluxes (≈758 and 690 L m^−2^ h^−1^ bar^−1^, respectively) and <2% rejection against the model pollutants sodium dodecylbenzene sulfonate (DBS) and dichloroacetic acid (DCA). When the membranes were submerged in the aqueous solutions and irradiated with UV-A LEDs, the photocatalytic performance factors for the degradation of DCA were similar to those obtained with suspended TiO_2_ particles (1.1-fold and 1.2-fold increase, respectively). However, when the aqueous solution permeated through the pores of the photocatalytic membrane, the performance factors and kinetics were two-fold higher than for the submerged membranes, mostly due to the enhanced contact between the pollutants and the membranes photocatalytic sites where reactive species were generated. These results confirm the advantages of working in a flow-through mode with submerged photocatalytic membranes for the treatment of water polluted with persistent organic molecules, thanks to the reduction in the mass transfer limitations.

## 1. Introduction

Clean water is an essential resource for sustainable development. However, the water supply is struggling to meet the increasing demand in many places [1]. Furthermore, climate change is expected to intensify water scarcity [2]. Accordingly, the importance of water reclamation and reuse is increasing. In addition, industrial developments have increased the variety of chemicals in water bodies. Some organic pollutants are toxic even at low concentrations and have low or no biodegradability, which limits the implementation of conventional biological treatments.

Photocatalysis is a promising alternative technology for various applications, including air and water purification, renewable energy production and materials’ synthesis [3]. With its potential to address global environmental challenges, photocatalysis has attracted significant research interest and investment in recent years. Special emphasis is placed on the removal of persistent organic pollutants from water, as this method is attractive because it can both degrade and mineralize dissolved organics compounds under ambient conditions without the addition of extra chemicals [4]. Photocatalysis is based on the activation of a semiconductor material when it is irradiated with light photons. These photons cause an electron (e^−^) to move from the valence band to the conduction band if the energy of the photons is greater than the band-gap of the semiconductor. Once the electron has been absorbed in the conduction band, a positive hole (h^+^) is formed on the valence band. When the electron/hole pair is produced, several reactive oxygen species are generated in the reaction medium, such as hydroxyl radicals or superoxide radicals, which are responsible for the degradation of the pollutants [5,6]. Titanium dioxide (TiO_2_) is a widely studied photocatalyst that has various advantages such as low toxicity, high stability and cost-effectiveness [7]. One of the major drawbacks of TiO_2_ is its wide band gap, ≈3.1–3.2 eV, which requires UV light for activation. Therefore, efforts have been made to develop visible light-active photocatalysts by modifying TiO_2_ or using other types of semiconductor materials [8,9,10,11,12,13,14,15,16]. Noble metals, such as Ag and Au, are commonly used to enhance the photocatalytic performance of TiO_2_, as they act as an electron sink. The formation of Schottky junctions between semiconductors and noble metal nanoparticles increases the efficiency of charge carrier separation [17,18].

In laboratory research, photocatalytic activity is often evaluated by mixing the powdery photocatalysts into the solution. The particles must therefore be removed from the treated water at the end of the experiment, by filtration, centrifugation or an external magnetic field in the presence of magnetic photocatalysts [19,20,21,22]. Alternatively, photocatalysts can be immobilised on a support, and then an extra facility to separate particles from water is not required [23]. One of the challenges of supported photocatalyst is the increase in mass transfer resistance, which can lead to slower kinetics. Another issue is the reactor design, as the total catalytic surface area per reactor volume tends to be smaller compared to a slurry-type reactor. Proper illumination plays a critical role in the removal of contaminants of emerging concern and the energy efficiency of photoreactors [24]. One possibility to assess whether the photocatalytic activity is improved is to work with a porous support, such as a porous membrane [25,26,27]. The use of ceramic membranes can be advantageous compared to organic polymeric membranes, overcoming the degradation of polymers under UV exposure [28,29]. Ceramic membranes typically have asymmetric structures with a thin separation layer and a macroporous support that provides the required mechanical strength. Geltmeyer et al. (2017) immobilised TiO_2_ nanoparticles on ceramic nanofibrous membranes by dip-coating for the photodegradation of the herbicide isoproturon [30]. Ahmad et al. (2020) coated TiO_2_ layer on a porous Al_2_O_3_ membrane substrate, developed using low-cost poly (vinyl chloride) (PVC) to generate interstitial voids for application in a photocatalytic membrane reactor for the treatment of organic dye contaminant. The photocatalytic activity of the membranes was evaluated using a batch photocatalytic reactor for the degradation of two dyes: acid orange (AO) and congo red (CR) [31]. Deepracha et al. (2021) deposited an active layer of commercial TiO_2_ powders (P25) on a microporous alumina membrane with pore size of 0.2 µm by the dip-coating method. This membrane was tested to examine the decomposition of phenol in water. The reaction was carried out using an easily up-scalable photocatalytic membrane reactor with four single-channel tubular membranes at transmembrane pressures of 250 mbar [32]. From another perspective, Singhapong et al. (2019) used porous mullites as ceramic membranes and coated them with TiO_2_ powders for disinfection applications. Uncoated mullites were used as control samples. The coating of TiO_2_ was able to inactivate *E. coli* under light exposure [33].

A combination of photocatalysis and membranes may realize synergetic performances, improving the contact between pollutants and photocatalyst. Very few studies have been carried out related to this topic. Albu et al. (2007) synthesised a dense and free-standing TiO_2_ nanotube membrane that was vertically oriented. Working in a flow-through operation, they reported the complete degradation of 2 × 10^−3^ mmol L^−1^ of the dye methylene blue after 4 h of UV irradiation [34]. Berger et al. (2020) evaluated the photocatalytic degradation of methylene blue. They operated under flow-through conditions with an aluminum oxide membrane coated with TiO_2_ via atomic layer deposition [35]. Regarding the degradation of volatile organic compounds, Moulis et al. (2015) described the oxidative degradation of acetone and methanol in gaseous phase in a flow-through photoreactor with immobilized TiO_2_. They focused their attention on the influence of the wavelength of UV light [36]. Presumido et al. (2022) presented a ceramic tubular membrane coated with a continuous graphene-TiO_2_ nanocomposite thin-film for the removal of contaminants of emerging concern in a single-pass flow-through operation [37]. They observed an important reduction in the fouling of the membranes during the filtration of the contaminated fluid when UV light was applied. Lofti et al. (2022) used a polyethersulfone-TiO_2_ membrane for the photocatalytic degradation of steroid hormone micropollutants in a continuous flow-through process [38]. The results indicate that the efficiency of the photocatalytic degradation of the micropollutants varied from 33% to 94%, depending on the type of hormone, concentration, and operating conditions.

In the present study, commercial TiO_2_ powder (P25, Evonik Industries, Essen, Germany) was immobilised on porous ceramic mullite tubes producing membranes with porosity in the range of microfiltration, in contrast to previous studies. Additionally, silver was photochemically deposited on some of the membranes. The permeation properties and photocatalytic performance of TiO_2_ and TiO_2_/Ag membranes were tested both with and without flow-through the membranes, all while under the influence of UV light irradiation. Two organic pollutants were chosen as examples of contaminants of emerging concern (CECs) in water: (i) sodium dodecylbenzene sulfonate (DBS), C_18_H_29_NaO_3_S, an important anionic surfactant used in the formulation of shampoos, detergents or cleaning products and, (ii) dichloroacetic acid (DCA), C_2_H_2_Cl_2_O_2_, a haloacetic acid used as fungicide and as a chemical intermediate in pharmaceuticals manufacture.

## 2. Materials and Methods

### 2.1. Materials

TiO_2_ (P25) photocatalyst (≈80% anatase and 20% rutile) with a specific surface area of approximately 55 m^2^ g^−1^ [39,40] was provided from Evonik Industries, Essen, Germany. Porous mullite tubes (outer diameter 12mm, inner diameter 9 mm, length 100 mm, mean pore size 1.3 μm, porosity 2%) were purchased from NIKKATO Corporation, Osaka, Japan. Silver acetate (CH_3_COOAg, purity 97%) was purchased from FUJIFILM Wako Pure Chemical Corporation, Japan. Sodium dodecylbenzene sulfonate (DBS) and dichloroacetic acid (DCA) were purchased from Sigma-Aldrich Chemie GmbH (Buchs, Switzerland) and Acros Organics (ThermoFisher Scientific, Geel, Belgium), respectively.

### 2.2. Methods

#### 2.2.1. Membrane Preparation and Characterisation

TiO_2_ particles were mechanically scrubbed onto the outer surface of the mullite tubes and heated in air at 673 K for three hours. After cooling down to room temperature, the membranes were washed with water to remove any loosely attached particles and dried at 353 K. Approximately 1 × 10^−2^ mmol cm^−2^ of TiO_2_ was immobilised on the membrane, calculated from the mass uptake after the deposition.

TiO_2_ membranes were immersed in 1 × 10^−2^ mmol L^−1^ silver acetate solution and exposed to UV light for 1.5 h. Black lamps (FL8BLB, Toshiba Corporation, Tokyo, Japan, λ_max_ = 352 nm, 3.3 mW cm^−2^) were used as the light source. Details can be found elsewhere [41]. During light irradiation, silver ions were photochemically reduced on the TiO_2_ surface [42]. The amount of silver deposited was calculated from the decrease in silver concentration in the solution, which was measured by inductively coupled plasma (ICP, SPS3500, S2 nanotechnology corporation, Yokohama, Japan). Approximately 3 × 10^−5^ mmol cm^−2^ of silver was deposited on the membrane. The membrane area was estimated to be about 29 cm^2^ for TiO_2_ membrane and 30 cm^2^ for TiO_2_/Ag membrane.

Characterisation of the membranes included scanning electron microscopy (FE-SEM, JSM-7600F, JEOL Ltd., Tokyo, Japan) and X-ray diffraction (XRD, SmartLab. Rigaku corporation, Tokyo, Japan).

#### 2.2.2. Permeation Tests

The permeation properties of the membranes were evaluated at room temperature (approximately 295 K) using either ultrapure water (MilliQ, Millipore, Burlington, VT, USA) or DBS model aqueous solutions. The experimental system used in the permeation tests is shown in Figure 1. The membrane was immersed in a liquid reservoir. One end of the membrane was blocked with a sealed plug and the other end was connected to a vacuum pump (Millivac Maxi SD1P014M04, Millipore, Darmstadt, Germany), which sucked the liquid from the reservoir through the membrane. A glass trap was connected between the membrane and the vacuum pump to collect the permeate, which was continuously monitored with a weight balance (PS 6000.R2, RADWAG, Spain) and recorded using Pomiar Win software. The transmembrane pressure ΔP was estimated as the difference between the atmospheric pressure (P_r_) and the vacuum pressure (P_v_) adjustable in the pump (Equation (1)).
(1)$$∆P=P_{r}-P_{v}$$

The membrane flux (L m^−2^ h^−1^), J, was calculated using Equation (2), where $Q_{permeate}$ is the permeate flux (L h^−1^) and $A_{e}$ (m^2^) is the effective membrane area:(2)$$J=\frac{Q_{permeate}}{A_{e}}$$

The hydraulic permeability (L m^−2^ h^−1^ bar^−1^), $K_{w}$, was estimated as indicated in Equation (3), considering the ultrapure water flux, $J_{w}$:(3)$$K_{w}=\frac{J_{w}}{∆P}$$

For the analysis of the permeability of organic model solutions, DBS (348.48 g mol^−1^) was chosen because of its large molecular size compared to DCA (128.94 g mol^−1^). A feed solution containing 50 mg L^−1^ of DBS was prepared. In addition to the flux of the model solution through the membrane samples, the compound rejection was estimated (Equation (4)):(4)$$R\left(\%\right)=\left[1-\frac{C_{0}}{C}\right]\times 100$$

After each use of the membrane, a cleaning protocol was followed, which consisted of washing the membrane several times with ultrapure (UP) water and allowing it to dry at room temperature.

#### 2.2.3. Photocatatlytic Activity Tests

Photocatalytic experiments were performed in a 1 L Pyrex glass photoreactor purchased to APRIA Systems SL (Cantabria, Spain). Figure 2 depicts the system. The equipment was provided with UV-A LED technology emitting at a fixed wavelength of 365 nm. The reactor housing had 30 LEDs (ENGIN LZ1-00UV00) distributed in 10 strips, 3 LEDs per strip, so as to homogeneously illuminate the entire height of the reactor. Strips were uniformly placed at a distance of 1.50 cm from the photoreactor. The solution was irradiated with an average irradiance of 200 W m^−2^. The reactor was placed on a magnetic stirrer to constantly mix the solution. The membrane was completely immersed in the aqueous feed solution. A lid, which ensured sufficient oxygenation of the aqueous medium, was placed to fix the membrane in the center of the vessel (see detail of the cross-section of the reactor in Figure 2).

First, dark adsorption experiments were carried out for 2 h. A total of 50 mg L^−1^ of DCA and DBS solutions were used with constant stirring. In order to evaluate the photocatalytic activity, the membranes were submerged inside the aqueous solutions. In this first approach, photocatalyst was immobilized on the membrane and operated without any permeation. The photocatalytic performance of the submerged membranes was then tested for 6 h at temperature and pH conditions similar to those used in the adsorption experiments.

In a second approach, coupled photocatalysis and permeation (flow-through mode) experiments were performed. In this case, the vacuum system used in the permeation tests was connected to the photocatalytic reactor. As a result, the aqueous solution was forced to permeate through the membrane, thus improving the solid–liquid contact between the photocatalyst and the pollutant solutions. The permeated volume, which had a total volume of 200 mL, was collected in the vacuum trap and subsequently returned to the feed tank at regular intervals of approximately 4–5 min. The total duration of the experiment was 6 h. The total flux through the membranes, the concentration of organic compounds in the feed tank and the accumulated permeate volume (4–5 min) were monitored.

All the experiments were carried out at room temperature (approximately at 295–298 K), natural pH (3.4 for DCA and 5.6 for DBS model solutions) and under constant stirring. All the experiments were performed with the same membrane samples.

The concentration of DCA was determined using an AS9-HC column in an ICS-5000 (Dionex, ThermoFisher Scientific, Waltham, MA, USA) ion chromatograph with a 9 mM Na_2_CO_3_ solution as eluent, at a flow rate of 1 mL min^−1^ and a pressure of approximately 2000 psi. The concentration of DBS was determined by UV spectrophotometry (UV-1800, Shimadzu Europe, Duisburg, Germany) at a wavelength of 223 nm. For UV spectrophotometry, a calibration curve was established with standard solutions up to 60 mg L^−1^.

## 3. Results

### 3.1. Characterisation

Figure 3 shows the membrane surface and a cross-sectional view. The outer surface of the ceramic tube was completely covered with TiO_2_ particles. The thickness of the TiO_2_ layer was approximately 2–3 μm. Some of the TiO_2_ membranes were immersed in a silver acetate solution and exposed to UV light. The concentration of silver in the solution decreased due to photoreduction, because silver ions were photochemically reduced on TiO_2_. The colour of the membrane changed from white to light yellow after UV irradiation. Under these conditions, silver can be deposited as metallic silver and silver oxide [43].

Figure 4 shows the XRD patterns of TiO_2_ particles (P25), mullite support, TiO_2_ membrane by immobilisation of P25 particles on mullite support and TiO_2_/Ag membrane. The P25 powder consists mainly of anatase phase with some rutile phase (≈80/20) [44]. The standard XRD pattern of TiO_2_ was checked through the Crystallographic Open Database (COD), card no. 7206075. The intensified diffraction peaks in the 2θ range at 25.4, 37.1, 48.14, 54.0, 55.2, 62.9, 68.7, 69.7 and 75.1° correspond to the (101), (004), (200), (105), (211), (204), (116), (220) and (215) crystallographic planes belonging to the anatase phase with tetragonal structure, respectively [45,46,47]. The peak at 27.4° is the characteristic reflection for rutile, whose crystallographic plane is (110) [48]. The uncoated mullites could be classified as microfiltration membranes [33]. The XRD pattern of mullite carrier at diffraction angles 2θ of 16.4, 26.1, 33.2, 35.3, 40.8° are assigned to (110), (210), (220), (111) and (121) crystallographic planes respectively [28]. Alumina phase is also detected in the sample, which could be explained by the transformation of mullite [49,50,51]. Peaks corresponding to mullite support and TiO_2_ were found in the TiO_2_ and TiO_2_/Ag membranes. The immobilisation of TiO_2_ on ceramic tubes did not significantly affect the TiO_2_ phases, since the annealing temperature used in this study, 673 K, is lower than the temperature that causes anatase to transition to rutile. The presence of silver was confirmed in earlier research through X-ray photoelectron spectroscopy (XPS) analysis [43]. Further characterisations of the membranes can be found in previous publications [41,43].

### 3.2. Permeation Performance

TiO_2_ and TiO_2_/Ag membranes showed an average hydraulic permeability of 758 ± 109 and 690 ± 96 L m^−2^ h^−1^ bar^−1^, respectively. The transmembrane pressure of the hydraulic permeability experiments varied from 0.95 to 1.07 bar. A similar hydraulic permeance (845 L m^−2^ h^−1^ bar^−1^) was reported by Ma et al. (2010) for porous α-Al_2_O_3_ ceramic disc supports coated with a hydroxyapatite (HPA) and TiO_2_/Ag layer [52]. The permeances of the DBS solutions were 611 ± 46 and 426 ± 63 L m^−2^ h^−1^ bar^−1^, for TiO_2_ and TiO_2_/Ag membranes, respectively. In both cases, the permeances for the DBS solutions are in the range of the hydraulic permeances of the TiO_2_ and TiO_2_/Ag membranes for water. Although the membrane permeance of the DBS solutions was generally slightly lower than the hydraulic permeance, none of the membranes presented permanent fouling and the membrane flux of all feed solutions was within the range of water permeation values. This is coherent with the low rejection of DBS (<2%) provided by the two types of membranes. Therefore, DBS was not retained by either TiO_2_ or TiO_2_/Ag membranes. Workneh and Shukla (2008) showed a rejection of sodium dodecyl sulfate in the range of 10–45% and a hydraulic permeability of 270 L m^−2^ h^−1^ bar^−1^ in a sodalite octahydrate-zeolite clay composite membrane [53]. Zhang et al. (2006) reported a rejection of 60–70% of DBS with a silica/titania nanorods/nanotubes composite membrane and a DBS solution permeation flux of 47 L m^−2^ h^−1^ at 0.5 bar ΔP [54]. When comparing results from the literature, it was observed that the membranes prepared in this work offer higher hydraulic permeabilities at the expense of a low rejection of organic molecules in the size range of DBS, confirming that the pore size of the tubular ceramic membranes remained in the microfiltration range after photocatalyst deposition.

In order to assess the stability and activity of the two membranes, TiO_2_ and TiO_2_/Ag, several interspersed cycles with water and DBS were carried out. These cycles involved the permeation of both water and DBS solutions through the membranes. After five consecutive cycles, the results showed no significant change in flux for either membrane when water or DBS solutions were permeated. This indicates that the activity of the membranes remained stable, and no loss of activity was detected during the cycles. These results provide valuable insights into the durability and performance of the TiO_2_ and TiO_2_/Ag membranes, indicating their potential usefulness for various applications in this field.

### 3.3. Photocatalytic Performance

TiO_2_ and TiO_2_/Ag membranes did not adsorb either DCA or DBS after 4 h of contact in the absence of light. Figure 5 shows the change in the dimensionless concentration of DCA and DBS during the photocatalytic tests using TiO_2_ and TiO_2_/Ag membranes. It could be observed that the TiO_2_/Ag membrane provided a slightly higher degradation percentage than TiO_2_ for both pollutants. The TiO_2_ membrane showed a degradation percentage of 16.6% for DBS and 37% for DCA after 6 h, whereas TiO_2_/Ag presented a degradation percentage of 21% for DBS and 44% for DCA. Experiments were carried out in duplicate with less than 5% error.

As observed, there is a slight increase in the removal of pollutants with the TiO_2_/Ag membrane, which could be attributed to the decrease in the electron/hole recombination rate due to the presence of small amounts of Ag, as Ag can act as an electron sink [9,55,56,57,58,59]. The electrons in the conduction band can be easily transferred to Ag, suppressing the recombination of the electron/hole pair, leading to more electrons becoming involved in the photocatalytic reaction. Therefore, the effective spatial separation of charge carriers was achieved, and led to the enhancement of the photocatalytic reaction [18]. The data fitted well to a pseudo-first-order kinetic model, typical for the degradation of organic pollutants by TiO_2_ photocatalysts, with 1.30 × 10^−3^ min^−1^ and 1.60 × 10^−3^ min^−1^ representing the kinetic constants corresponding to DCA degradation, and 5.02 × 10^−4^ min^−1^ and 6.27 × 10^−4^ min^−1^ representing the kinetic constants for DBS degradation, respectively, for the membranes without and with Ag.

Figure 6 presents the total flux and the dimensionless DCA concentration decay during the simultaneous photocatalysis and flow-through permeation experiments with TiO_2_ (Figure 6a) and TiO_2_/Ag membranes (Figure 6b). It also compares the DCA concentration decay in the photocatalytic experiments without permeation with the photocatalytic material supported on the membrane. Experimental results are shown with error bars; the average standard deviation was lower than 5% for all samples.

A negligible loss of activity with these operation cycles was previously confirmed. Furthermore, the DCA concentration in the tank and in the collected permeate were the same each time, also confirming the null DCA rejection of TiO_2_ and TiO_2_/Ag membranes. Interestingly, when the feed solution permeated the membrane pores, the DCA degradation rate was increased. For the TiO_2_ membrane, a kinetic constant *k* of 2.50 × 10^−3^ min^−1^ (R^2^ = 0.99) was obtained, which is almost two-fold higher than that obtained when working with the submerged membrane. The TiO_2_/Ag membrane operating in a flow-through mode provided a value of *k* of 2.80 × 10^−3^ min^−1^ (R^2^ = 0.99), which is 1.8-fold higher than the kinetic constant corresponding to the submerged membrane without permeation. These results confirm that as the aqueous solution permeated the pores of the membrane, the contact between the pollutants and the oxidant species generated on the surface of the TiO_2_ and TiO_2_/Ag membranes was facilitated, which otherwise could not react with the pollutant molecules due to the mass transport limitations [38]. The flow-through mode, therefore, enhanced the degradation rate of the photocatalytic process and is emerging as a promising option [34,35,36,37,38,60]. Again, the results of the TiO_2_/Ag membrane are slightly better than those of the TiO_2_ membrane, due to the positive effect of Ag as an electron sink [61,62]. This confirms that the presence of Ag, even in small amounts, has the ability to improve the photocatalytic results. Bian et al. (2013) reported a strategy for loading uniformly Ag nanoparticles in flow-through TiO_2_ nanotube arrays [61]. They tested the nanotubes arrays in the degradation of methyl orange under UV light, reaching complete degradation in 30 min. They attributed the good results to the good distribution of silver in the nanotube arrays. Working in a flow-through operation mode significantly attenuates the difference in the photocatalytic performance between TiO_2_ and TiO_2_/Ag membranes. Furthermore, in economic terms, it does not compensate for the presence of silver in the preparation of the materials.

For comparison, DCA degradation experiments with commercial TiO_2_ in suspension were carried out in the same photocatalytic reactor. To facilitate the comparison between suspended and immobilised TiO_2_, a performance factor was calculated as the ratio of initial degradation rate and catalyst concentration (Table 1).

An analysis of the results in Table 1 showed that the performance factor was similar (1.1-fold higher) when TiO_2_ was used in suspension and for the submerged membrane, indicating that, in this case, the immobilised photocatalyst preserved the photocatalytic activity compared to slurry systems. Interestingly, the performance factor of the immobilised photocatalysts operating with the submerged membrane in flow-through mode was 2.2-fold higher than with TiO_2_ in suspension. The membranes did not retain the pollutants, so the increase in the kinetic constants could be attributed to the improved contact between the pollutants and the oxidative species favored by the transport through the pores of the membrane. Therefore, these results demonstrate that the performance of photocatalytic membranes for the degradation of aqueous pollutants is enhanced when working in a flow-through mode because the contact with reactive species is favored and the resistance to mass transport is reduced, leading to an increase in the overall degradation rate.

## 4. Conclusions

Water remediation is crucial to environmental preservation, removing organic compounds from water sources. The use of combined technologies is an interesting approach for this purpose. In this work, mullite tubular membranes were employed as supports for immobilised TiO_2_ and TiO_2_/Ag. These membranes were used to evaluate the hydraulic permeability, with values of ≈758 and 690 L m^−^^2^ h^−^^1^ bar^−^^1^ obtained for TiO_2_ and TiO_2_/Ag membranes, respectively. The permeabilities for DBS solutions were in the range of the hydraulic permeabilities of the membranes. In addition, neither TiO_2_ nor TiO_2_/Ag membranes retained DBS. Afterwards, the same membranes were tested for the photocatalytic degradation of DCA and DBS synthetic solutions. Although silver improves the photocatalytic properties, the improvement achieved under the conditions of this work did not support the large-scale manufacture of silver-containing membranes. Finally, the combination of both technologies, filtration and photocatalysis, operating in a single flow-through mode for the treatment of synthetic solutions activated by UV-A light improves the process’ effectiveness. Working in flow-through mode with the TiO_2_ membrane improves the performance factor by 2.2-fold compared to suspended TiO_2_. This improvement is directly related to the enhanced contact between the organic molecules and the oxidative species generated on the surface of the membrane under UV-A light, and reinforces one of the main advantages of immobilised membranes, facilitating the catalyst recovery after the photocatalytic treatment. The results of this study indicate that working with the submerged membrane in flow-through mode is a promising approach for practical environmental applications.

## Figures and Tables

**Figure 1 membranes-13-00448-f001:**
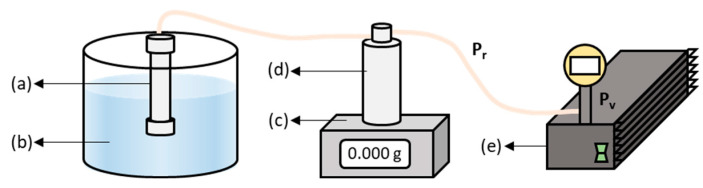
Permeability system consisting on: (**a**) membrane, (**b**) solution reservoir, (**c**) weight balance, (**d**) vacuum trap and (**e**) vacuum pump.

**Figure 2 membranes-13-00448-f002:**
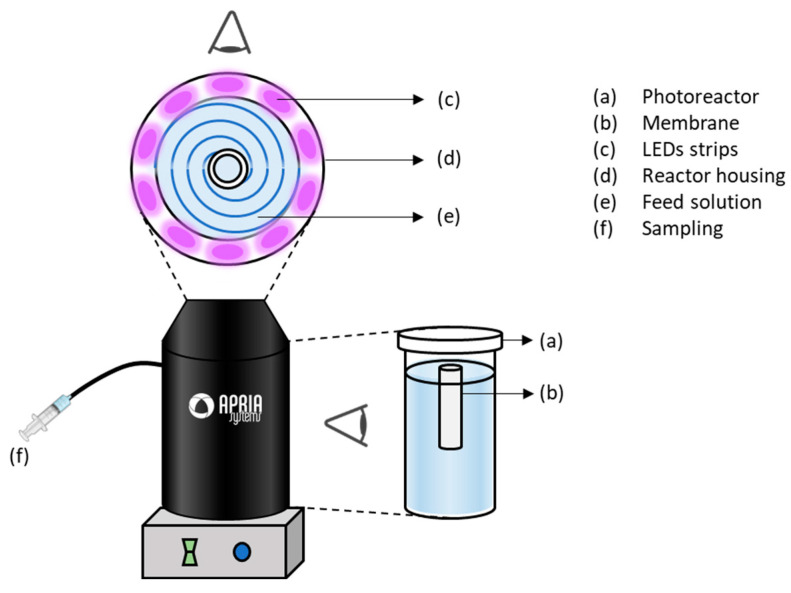
Schematics of the photocatalytic system.

**Figure 3 membranes-13-00448-f003:**
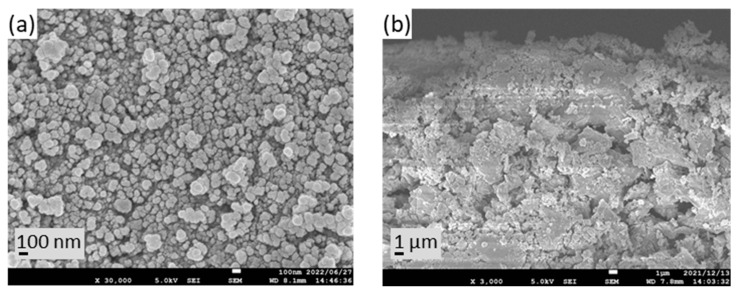
TiO_2_ membrane morphology: (**a**) surface view, (**b**) cross-sectional view.

**Figure 4 membranes-13-00448-f004:**
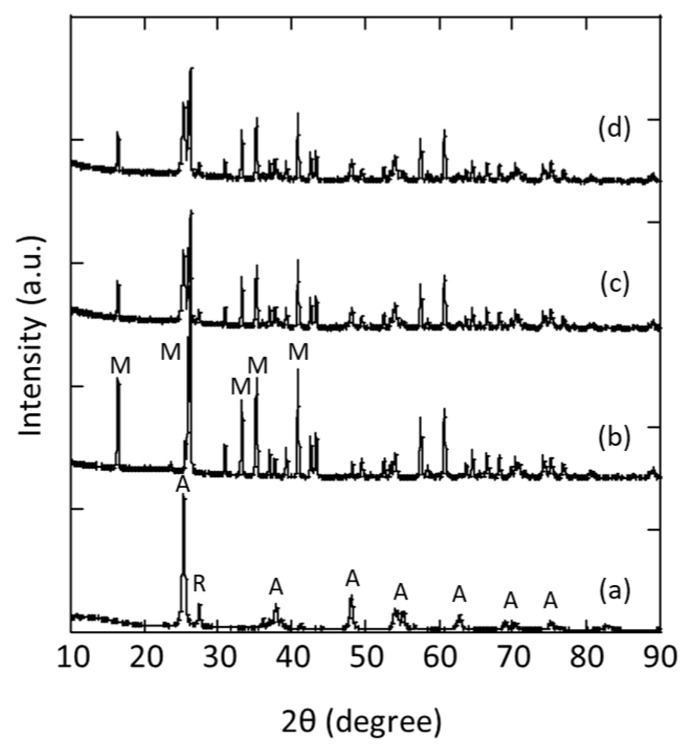
XRD patterns: (**a**) P25 powder, (**b**) mullite support, (**c**) TiO_2_ membrane, (**d**) TiO_2_/Ag membrane.

**Figure 5 membranes-13-00448-f005:**
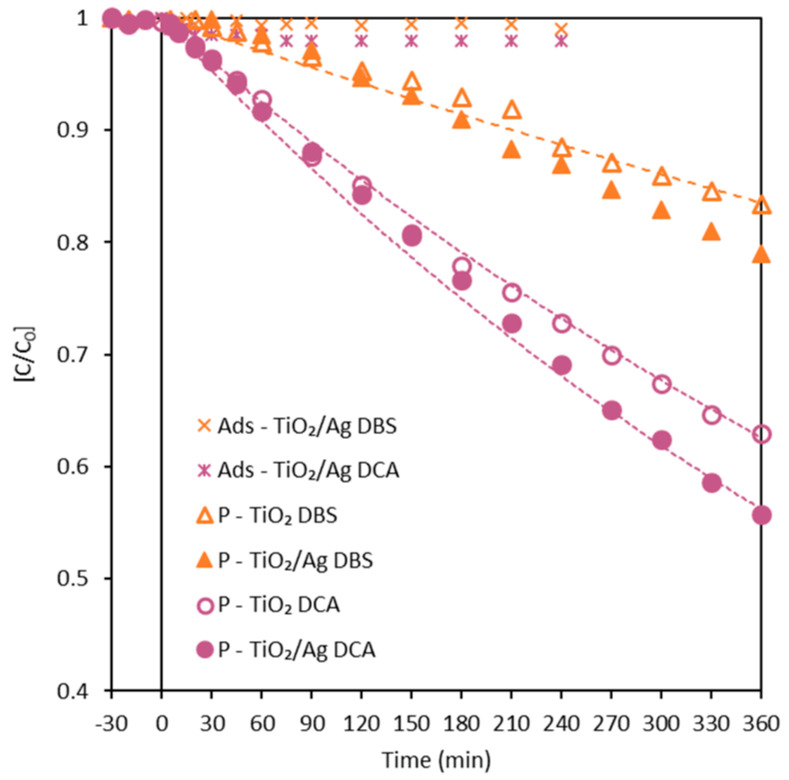
Photocatalytic degradation data of DBS and DCA with TiO_2_ and TiO_2_/Ag submerged membranes without permeation. Dotted lines represent the fitted curves to pseudo-first-order kinetic model. P: photocatalytic experiments.

**Figure 6 membranes-13-00448-f006:**
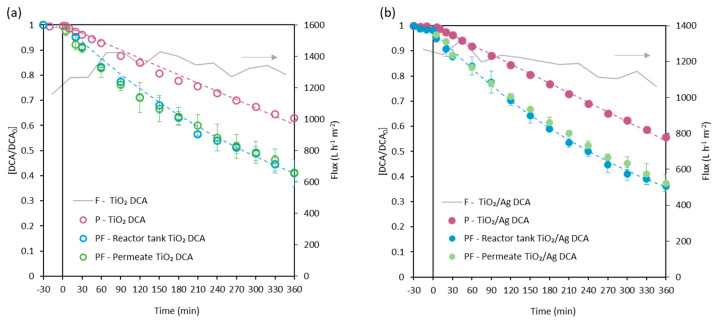
Comparison of DCA degradation in photocatalytic submerged membranes (P: photocatalytic experiments) and coupled photocatalysis and flow-through permeation experiments (PF: photocatalytic and flow-through). Total flux, in grey lines, of flow-through permeation experiments at a transmembrane pressure of 1 bar. (**a**) TiO_2_ membrane and (**b**) TiO_2_/Ag membrane.

**Table 1 membranes-13-00448-t001:** Comparison of the photocatalytic performance of TiO_2_-based catalysts: suspended, P: submerged photocatalytic membrane, PF: photocatalytic membrane and flow-through permeation experiments for DCA degradation.

Photocatalytic Experiment	[TiO_2_] (g L^−1^)	r_0_ (mg L^−1^ min^−1^)	Performance Factor (mg L^−1^ min^−1^ g_cat_^−1^)
TiO_2_-P25 in suspension	0.30	0.79 ± (2.79 × 10^−2^)	2.63 ± 0.11
P-TiO_2_	0.02	0.06 ± (1.16 × 10^−3^)	3.01 ± 0.12
PF-TiO_2_	0.02	0.12 ± (4.13 × 10^−3^)	5.80 ± 0.24

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
