# Peer review of "Performance of TiO2-Based Tubular Membranes in the Photocatalytic Degradation of Organic Compounds"

_membranes, 2023, doi:10.3390/membranes13040448_

Round 1

Reviewer 1 Report

This articles deals with composite ceramic membranes on which TiO2 and TiO2/Ag photocatalysts were immobilized for a photocatalytic degradation of organic pollutants present in water. The study aims to highlight, using 2 model pollutants DBS and DCA, that the solution permeation improves the photocatalytic degradation (compared with submerged membranes) because of reduced mass transfer limitations.

Although the problem addressed is interesting, the introduction does not position the study clearly enough in relation to the state of the art:

-          what work has already been done on the immobilization of photocatalysts on porous ceramic membranes, what is their implementation (closed or continuous reactor) and the photocatalytic performances obtained?

-          the passage on the improvement of the oxygen supply is not clearly formulated

-          it will also be necessary to identify more clearly the expectations regarding the use of a TiO2/Ag combination compared to that of TiO2 alone (especially since in some results, such as those presented in Figure 6, the interest of money is not obvious)

Regarding the Material and Methods paragraph:

-          it is noted that the permeation tests are performed at "room temperature". Rigorously, it would be advisable to correct the flow in relation to the variation of viscosity with temperature.

-          Photocatalytic tests are performed with a LED strip positioned above the solution in which the 10 cm long membrane is immersed; is it certain that the TiO2 deposited on the part closest to the bottom of the reactor receives the same irradiation as that deposited on the upper part of the membrane?

Results:

-          What is the effect of photocatalysts deposit on membrane porosity ? were permeation tests performed on porous mullite support as a reference for permeation values (in addition to the comparison of DCA contents in feed solutions and permeate) ?

-          the authors write that the tested membranes did not show any permanent fouling phenomenon (l.201): were they used over a longer period than the 5 cycles of 6h? were the permeability values before/after DBS filtration over 5 cycles compared (there is only a qualitative appreciation l.213) ? insofar as the same membranes are used for the different tests, is a cleaning protocol set up between the tests?

-          If adsorption tests didn’t show any removal of DBS over 2h -6h would have been better-, these results should appear on figure 5 as “reference”.

-          to confirm the contribution of the continuous mode, through a facilitation of the transfer, were the tests in static reactor in which the membrane is immersed carried out at various speeds of agitation?

Reviewer 2 Report

Referee report: “Performance of TiO2-based tubular membranes in the photocatalytic degradation of organic compounds

This is a quite interesting article that probably can be recommended for publication, but only after clarifying and detailing some parts of the text.

1.     Line 42.  After [5-10] , which are already quite old , a few more recent MDPI articles on this subject can be mentioned here:

Tsebriienko, T.; Popov, A.I. Effect of Poly(Titanium Oxide) on the Viscoelastic and Thermophysical Properties of Interpenetrating Polymer Networks. Crystals 202111, 794. https://doi.org/10.3390/cryst11070794

Trestsova, M.A.; et al. Oxidative C-H/C-H Coupling of Dipyrromethanes with Azines by TiO2-Based Photocatalytic System. Synthesis of New BODIPY Dyes and Their Photophysical and Electrochemical Properties. Molecules 202126, 5549. https://doi.org/10.3390/molecules26185549

Dorosheva, I.B., Valeeva, A.A., Rempel, A.A. et al. Synthesis and Physicochemical Properties of Nanostructured TiO2 with Enhanced Photocatalytic Activity. Inorg Mater 57, 503–510 (2021). https://doi.org/10.1134/S0020168521050022

2.     Line 62. What structure do TiO2 powders have?

3.     Fig.3. The quality of the drawing needs improvement, the legend is visible.

4.     Lines 216-222.  What is the physical reason for such behavior?

5.     Have any radiation aging effects been observed in after X-ray diffraction measurement?

6.     Data in Table 1 need corresponding error bars.

7.     Line 231. Does this mean that silver ions are centers for trapping electrons and holes?

8.     In the conclusions, it is necessary to formulate more clearly what new data on the studied materials were obtained in this work?

In general, the manuscript is interesting and can be recommended for publication after constructive reflection on the above comments.

Round 2

Reviewer 1 Report

the comments have been taken into account to improve the paper

Reviewer 2 Report

Аfter detailed revision this manuscript can be

recommended for publication.